# Dual RNA-Seq Profiling Unveils Mycoparasitic Activities of *Trichoderma atroviride* against Haploid *Armillaria ostoyae* in Antagonistic Interaction Assays

Liqiong Chen,[a] Simang Champramary,[a,b] Neha Sahu,[c] Boris Indic,[b] Attila Szűcs,[a] Gábor Nagy,[a] Gergely Maróti,[d] Bernadett Pap,[e] Omar Languar,[a,b] Csaba Vágvölgyi,[a] László G. Nagy,[c] László Kredics,[a] György Sipos[b]

[a]Department of Microbiology, Faculty of Science and Informatics, University of Szeged, Szeged, Hungary
[b]Functional Genomics and Bioinformatics Group, Institute of Forest and Natural Resource Management, Faculty of Forestry, University of Sopron, Sopron, Hungary
[c]Synthetic and Systems Biology Unit, Biological Research Center, Szeged, Hungary
[d]SeqOmics Biotechnology Ltd., Mórahalom, Hungary
[e]Institute of Plant Biology, Biological Research Center, Szeged, Hungary

Liqiong Chen and Simang Champramary contributed equally to this work. Author order was determined on the basis of seniority.

**ABSTRACT** *Armillaria ostoyae*, a species among the destructive forest pathogens from the genus *Armillaria*, causes root rot disease on woody plants worldwide. Efficient control measures to limit the growth and impact of this severe underground pathogen are under investigation. In a previous study, a new soilborne fungal isolate, *Trichoderma atroviride* SZMC 24276 (TA), exhibited high antagonistic efficacy, which suggested that it could be utilized as a biocontrol agent. The dual culture assay results indicated that the haploid *A. ostoyae*-derivative SZMC 23085 (AO) (C18/9) is highly susceptible to the mycelial invasion of TA. In the present study, we analyzed the transcriptome of AO and that of TA in *in vitro* dual culture assays to test the molecular arsenal of *Trichoderma* antagonism and the defense mechanisms of *Armillaria*. We conducted time-course analysis and functional annotation and analyzed enriched pathways and differentially expressed genes including biocontrol-related candidate genes from TA and defense-related candidate genes from AO. The results indicated that TA deployed several biocontrol mechanisms when confronted with AO. In response, AO initiated multiple defense mechanisms to protect against the fungal attack. To our knowledge, the present study offers the first transcriptome analysis of a biocontrol fungus attacking AO. Overall, this study provides insights that aid the further exploration of plant pathogen-biocontrol agent interaction mechanisms.

**IMPORTANCE** *Armillaria* species can survive for decades in the soil on dead woody debris, develop rapidly under favorable conditions, and harmfully infect newly planted forests. Our previous study found *Trichoderma atroviride* to be highly effective in controlling *Armillaria* growth; therefore, our current work explored the molecular mechanisms that might play a key role in *Trichoderma-Armillaria* interactions. Direct confrontation assays combined with time course-based dual transcriptome analysis provided a reliable system for uncovering the interactive molecular dynamics between the fungal plant pathogen and its mycoparasitic partner. Furthermore, using a haploid *Armillaria* isolate allowed us to survey the deadly prey-invading activities of the mycoparasite and the ultimate defensive strategies of its prey. Our current study provides detailed insights into the essential genes and mechanisms involved in *Armillaria* defense against *Trichoderma* and the genes potentially involved in the efficiency of *Trichoderma* to control *Armillaria*. In addition, using a sensitive haploid *Armillaria* strain (C18/9), with its complete genome data already available, also offers the opportunity to test possible variable molecular responses of *Armillaria ostoyae* toward diverse *Trichoderma* isolates with various biocontrol

Address correspondence to György Sipos, sipos.gyorgy@uni-sopron.hu, or László Kredics, kredics@bio.u-szeged.hu.

The authors declare no conflict of interest.

abilities. Initial molecular tests of the dual interactions may soon help to develop a targeted biocontrol intervention with mycoparasites against plant pathogens.

**KEYWORDS** dual culture assay, root rot disease, biocontrol-related candidate genes, transcriptome, *Armillaria ostoyae*, *Trichoderma atroviride*, time-course analysis, defense-related candidate genes

A*rmillaria* species rank among the most damaging plant-pathogenic fungi and are best known to cause root rot diseases in a broad spectrum of woody plants (1). Infections by *Armillaria* species can have severe consequences in forest ecosystems, including plant fitness reduction and substantial tree mortality (2). *Armillaria ostoyae* is a facultative necrotrophic species that prefers coniferous habitats and invades coniferous trees as a primary pathogen. *Armillaria* infections are soilborne, and they are spread by rhizomorphs, as well as through root-to-root contacts. After entering the plant, the invading mycelial fans of *Armillaria* paralyze the root vascular tissues, causing nutrient uptake failure and dramatic vigor decrease, leading to the instant mortality of infected trees (3). Although root infections kill the tree, the fungus colonizes the root system saprophytically and uses it as a food source to attack adjacent healthy trees. Inoculum removal of the infected trees is not an effective way to clean the affected area from *Armillaria* rhizomorphs and mycelia, as most of the rhizomorphs extend deep into the soil, wherefrom they can subsequently colonize newly planted trees (4). Hence, soil infestation caused by *Armillaria* can last for decades and is extremely difficult to eradicate (3, 4).

The severe economic losses and ecological damage triggered by pathogenic *Armillaria* species require robust control strategies. Unfortunately, standard silvicultural control treatments and fungicide-based chemical control measures have many drawbacks (5). Nevertheless, biocontrol strategies, in which beneficial microorganisms help to prevent plant diseases, offer a potentially better option (6). Biocontrol measures emphasize environmental protection and ecological sustainability by harnessing beneficial microorganisms, like free-living soil fungi from the genus *Trichoderma* (7). This strategy is conducive to animal and human safety and benefits the environment, as there are no pesticide residues. Furthermore, most biocontrol agents (BCAs) effectively protect plants from various pathogen-induced diseases and offer the potential to promote plant growth, resulting in increased harvests (8).

Species from the genus *Trichoderma* exhibit high adaptability and survival potential; therefore, they are widespread in soil. Their biocontrol mechanisms rely on multiple modes of action such as mycoparasitic activities on plant-pathogenic fungi (9), competition for living space and limited nutrients (10), antibiotic effects on the target pathogens (11), induction of plant resistance to pathogens, and plant growth promotion (12). The inhibitory effect of *Trichoderma* BCAs on *Armillaria* pathogens has been extensively studied (13). One report demonstrated that *Trichoderma* species adapt quickly according to the actual habitat and overgrow *Armillaria* colonies by competing for limited nutrient resources (14). Selected *Trichoderma* isolates successfully restrained pathogen growth through indirect interactions of volatile and nonvolatile metabolites and directly by mycoparasitic interactions to degenerate and lyse the mycelia of various plant-pathogenic fungi (15). In a previous study, we isolated and identified 14 *Trichoderma* species from forest soils (16). Most of these species rapidly occupied culture space, gradually produced large amounts of conidia on the mycelial surface of *Armillaria* colonies, and potentially inhibited the growth of 4 tested *Armillaria* species, namely, *A. ostoyae*, *A. mellea*, *A. cepistipes*, and *A. gallica*. Biocontrol capabilities make *Trichoderma* species important organisms for further research at the genome or transcriptome level to search for specific or novel genes associated with mycoparasitism, biodegradation, signaling, and other invasive microbial activities.

To date, the molecular repertoire of *T. virens* required for antagonism during its interaction with *Rhizoctonia solani* has been reported (17); however, the transcriptome-

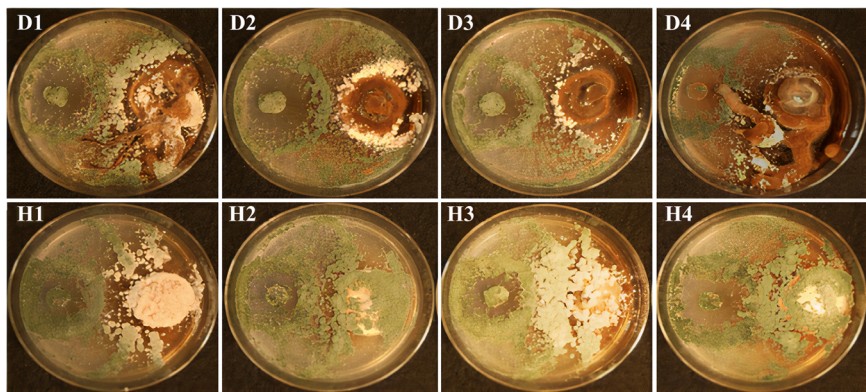

**FIG 1** Antagonistic effects of *T. atroviride* against various diploid (D1–D4) and haploid (H1–H4) strains of *A. ostoyae* in a dual culture assay on PDA medium. AO and TA strains were inoculated to the right and left side of the plates, respectively, as described in section "*In vitro* antagonistic activity assessment by dual culture assay" of Materials and Methods. Melanized rhizomorphs were developed on the surface of the medium in the case of all diploid strains, while haploid AO strains did not produce any rhizomorphs and were easily overgrown by TA, suggesting that they are more suitable for studying pure hyphal interactions of the two fungi and the mycoparasitic attack of TA on AO.

level interactions between *Trichoderma* and *Armillaria* have not yet been studied. This study provides the first insights into the biocontrol interaction-induced transcriptomes of *Armillaria* and *Trichoderma*, which may help to improve our understanding of the molecular interaction mechanisms, thereby offering guidance for biofungicide development and employment.

## RESULTS

**Antagonistic effect of *T. atroviride* SZMC 24276 against *A. ostoyae* strains.** The *T. atroviride* SZMC 24276 strain was selected as a BCA based on its outstanding antagonistic activity in our previous experiments (biocontrol index values >80 for more than 10 examined *Armillaria* strains from 4 species *A. cepistipes*, *A. ostoyae*, *A. gallica*, and *A. mellea*) (16). In the present study, all diploid strains of *A. ostoyae* displayed typical *Armillaria* morphology with abundant rhizomorph formation and mycelial growth on potato dextrose agar (PDA). Aerial hyphae of the diploid strains could be differentiated by their morphological attributes; they appeared to darken and harden. In contrast, the mycelia of the haploid derivatives of *A. ostoyae* grew white and fluffy, and they continued producing whitish homogeneous abundant aerial mycelia. During the 105-h coincubation of dual cultures, *T. atroviride* SZMC 24276 showed antagonistic effects against diverse diploid and haploid strains of *A. ostoyae* (Fig. 1; Table 1). *T. atroviride* grew rapidly toward the colony of *A. ostoyae* and gradually invaded the growth area of *A. ostoyae* strains. The results revealed that by the fifth day, *Trichoderma* easily overgrew haploid *Armillaria* isolates as confirmed by the abundant green conidia on the surface of the haploid *Armillaria* mycelia compared to that on diploid strains. For these reasons, the *A ostoyae*-haploid derivative SZMC 23085 (AO) was selected for the transcriptome analysis of its interaction with *T. atroviride* SZMC 24276 (TA).

**Time-course analysis to understand the interaction dynamics between *T. atroviride* and *A. ostoyae*.** We employed the dual culture method to study the interaction between TA and AO at three different time points: 53, 62, and 105 h after the inoculation

**TABLE 1** Biocontrol index (BCI) values of *T. atroviride* SZMC 24276 in the dual culture interaction tests with various diploid and haploid *A. ostoyae* isolates

| | Diploid | | | | Haploid | | | |
|---|---|---|---|---|---|---|---|---|
| **Test** | **SZMC 23083 (D1)** | **SZMC 24127 (D2)** | **SZMC 24128 (D3)** | **SZMC 24129 (D4)** | **SZMC 27047 (H1)** | **SZMC 27048 (H2)** | **SZMC 27049 (H3)** | **SZMC 23085 (H4)** |
| BCI | 88.13 ± 0.56 | 89.26 ± 1.62 | 93.29 ± 2.99 | 85.49 ± 8.03 | 99.40 ± 1.04 | 99.55 ± 0.77 | 99.37 ± 1.09 | 98.46 ± 1.83 |

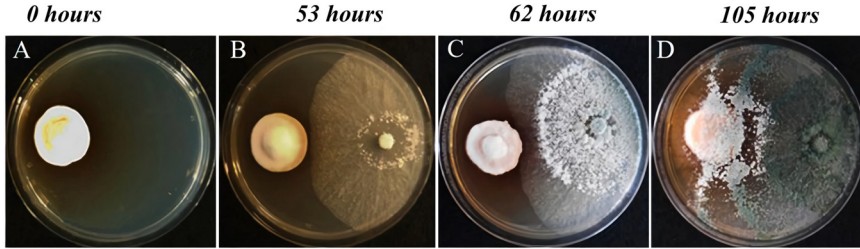

**FIG 2** *In vitro* interaction between *A. ostoyae and T. atroviride* at different time points. Individually growing, noninteracting haploid *A. ostoyae* culture (A). Interacting cultures of *A. ostoyae* and *T. atroviride* at 53 h (B), 62 h (C), and 105 h (D).

of *T. atroviride* (Fig. 2). RNA sequencing (RNA-Seq) analysis of coscrapped TA-AO samples at 105 h did not show any transcripts for AO (0.3% of reads could be mapped to *A. ostoyae*) (Fig. S1 in the supplemental material). Since we focused on understanding the mutual metabolite-level and mycoparasitic interactions, we did not consider the 105-h interaction sample in this study. On average, 17 million reads with a mean read length of 150 bp were obtained for both TA and AO after quality trimming. We performed a time-course analysis of the transcriptome data and generated three significant clusters for TA and AO (Fig. 3). From the clusters, we identified the genes showing the highest expression in the metabolite and mycoparasite stages of the interaction, along with the genes displaying a continuous downtrend pattern in TA and AO (Table 2).

**Gene expression profiling of the major trends.** To analyze major gene expression trends, we considered TA-AO genes showing continuous downtrend patterns or genes exhibiting the highest upregulation at the metabolite or mycoparasite stages of the interaction (Fig. 3; Table 2) for further analysis.

**Downtrend genes in *A. ostoyae* and *T. atroviride*.** We observed 768 and 747 downtrend genes in AO and TA, respectively (Table 2; Fig. 3; Supplemental File S1). GO enrichment analysis of AO downtrend genes showed the enrichment of 14 biological

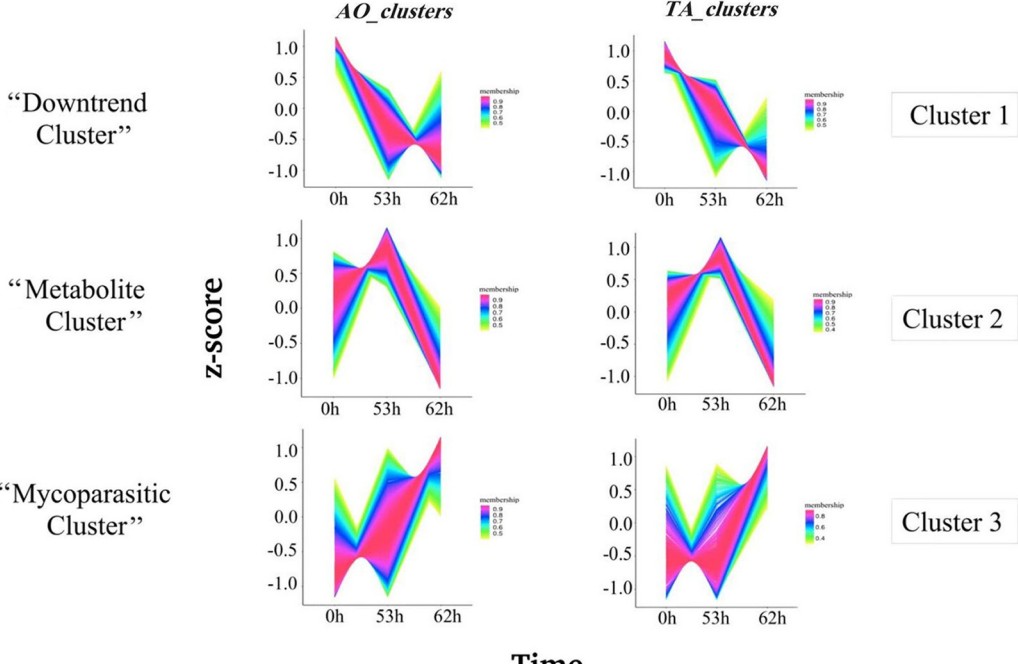

**FIG 3** Cluster analysis of genes based on fold change after 0, 53, and 62 h. Cluster analysis of genes was performed using TCseq. The *x* axis denotes the different time points of *in vitro* interaction, and the *y* axis represents the $\log_2$-fold change. Members of the "Downtrend cluster" have shown downregulation since the interaction started. The "Metabolite cluster" genes showed the highest upregulation during the 53rd hour, and the genes of the "Mycoparasite cluster" showed the highest upregulation at the 62nd hour.

**TABLE 2** Total number of genes grouped by cluster according to the stages where they show differential regulation

| Organism | Metabolite stage cluster genes | Mycoparasite stage cluster genes | Downtrend genes |
|---|---|---|---|
| AO | 250 | 1,412 | 768 |
| TA | 1,024 | 985 | 747 |

processes (BPs), 3 molecular functions (MFs), and 20 cellular components (CCs) (Fig. 4). Enriched biological processes in AO included cell cycle, DNA replication, DNA-dependent DNA replication, DNA repair, mitotic cell cycle, sterol metabolic/biosynthetic processes, and cellular response to stress/DNA damage stimulus (Fig. 4A and B). However, TA exhibited the enrichment of 13 BPs, 2 MFs, and 5 CCs (Supplemental File S1). Biological processes enriched in TA included oxidation-reduction activity and the metabolic processes of amino acids, organic acids, cellular ketones, and sulfur (Fig. S2).

**The metabolite and mycoparasite stages of *Armillaria-Trichoderma* interaction. (i) InterPro enrichment.** During the metabolite stage, 250 AO genes and 1,024 TA genes (Table 2; Supplemental File S2) were upregulated exclusively. Functional analysis of these genes from AO showed enrichment of InterPro domains such as cyanate lyase, glutathione peroxidase, SnoaL, indoleamine 2,3-dioxygenase, phenol hydroxylase, nonribosomal peptide synthetase (NRPS) condensation domain, flavin-dependent halogenase, Ctr copper transporter, DAHP synthetase, and class I-like SAM-dependent *O*-methyltransferase (Fig. 5A). TA showed enrichment of cerato-ulmin (hydrophobin), chaperonin Cpn60, nucleoside phosphorylase, NACHT nucleoside triphosphatase, glycoside hydrolase 3, NRPS condensation domain, heterokaryon incompatibility, and ankyrin repeat-containing domain (Fig. 5B).

In the mycoparasite stage, we identified 985 genes in TA and 1,412 genes in AO that were specifically upregulated (Supplemental File S2). During this stage, gene families such as Pex (peroxisomal biogenesis), Hsp20, CDR ATP-binding cassette (ABC) transporters, thiolase, NAD(P) binding domain, cytochrome $b_5$-like heme binding, and short-chain dehydrogenases/reductases (SDRs) were enriched in TA (Fig. 5D). AO exhibited upregulation of domains such as beta-1,4-cellobiohydrolase, DSBA-like thioredoxin, P-type ATPase, NmrA, malic acid transport protein, DJ-1, DUF2235, NADH:flavin oxidoreductase, and voltage-dependent anion channel (Fig. 5C).

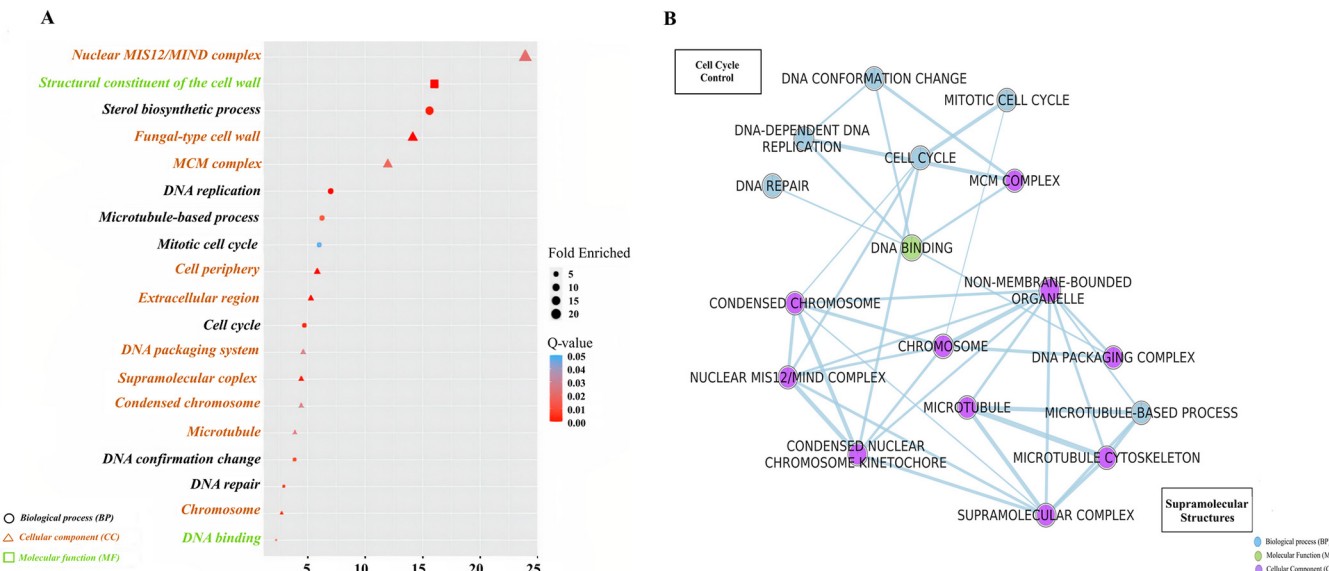

**FIG 4** Gene ontology (GO) enrichment analysis of *A. ostoyae* downtrend cluster. (A) GO dot plot of biological processes (BP), cellular component (CC), and molecular function (MF) enriched in the downtrend cluster. The size and the color of the dots represent the fold enriched and the q value, respectively. The shape and color of the GO terms in the *y* axis denote the GO type. (B) GO enrichment map of significant BP, CC, and MF terms generated using Enrichment map in Cytoscape v3.8.2. Blue, green, and purple colored nodes represent the type of GO, and the edges denote gene overlaps. The size of the edge resembles the size of overlaps.

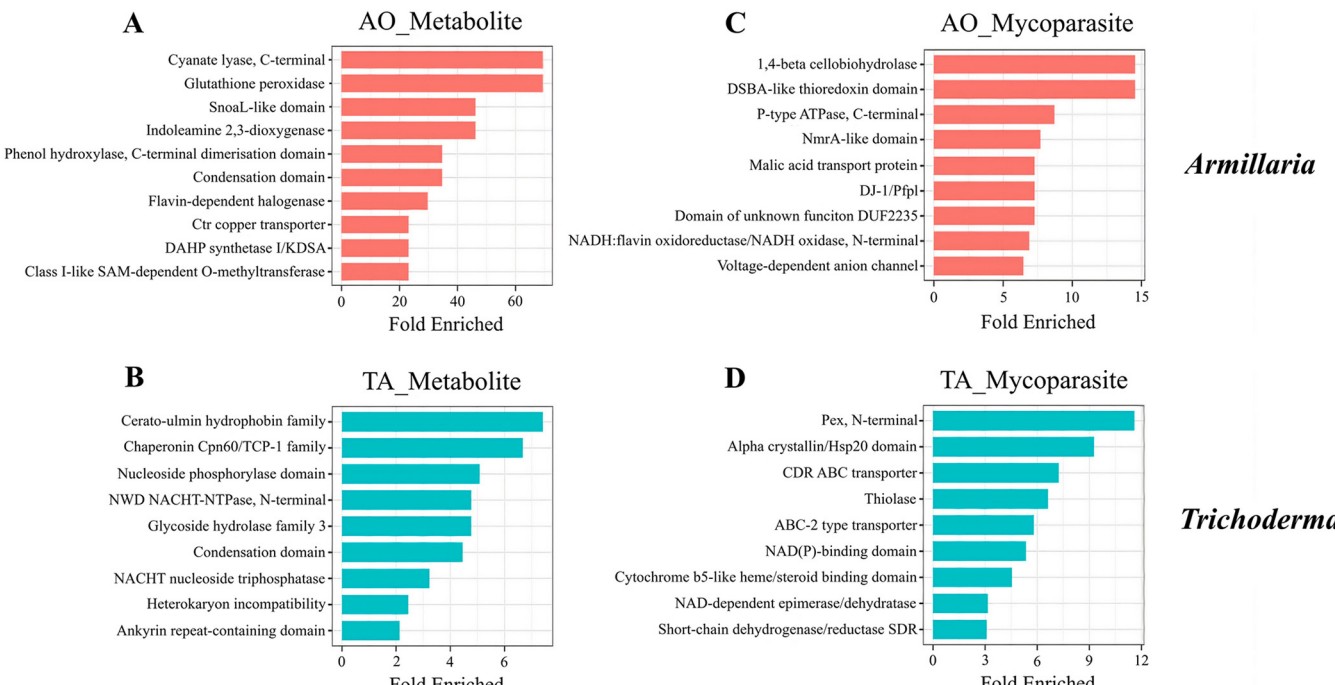

**FIG 5** InterPro enrichment analysis of the metabolite and mycoparasite clusters in *A. ostoyae* and *T. atroviride*. Bar plot of top InterPro families enriched (*P* < 0.05) is shown in *A. ostoyae* (A and C) and *T. atroviride* (B and D) during the metabolite (A and B) and mycoparasite (C and D) stages of interaction. The *y* axis denotes the InterPro families, and the *x* axis represents the fold enriched.

**(ii) Gene ontology enrichment analysis.** Gene ontology (GO) enrichment analysis of genes during the metabolite stage of the interaction showed enrichment of the indole to kynurenine, small molecule, organic acid, aromatic, and alpha-amino acid metabolic processes in AO (Fig. 6A and B; Supplemental File S1), and the ergosterol and cellular alcohol biosynthetic, as well as the carbohydrate metabolic processes (Fig. S3), in TA. Then, during the direct mycoparasitic interaction phase, AO upregulated indole alkaloid biosynthesis, cellular detoxification, and various transport functions related to C4 dicarboxylate, malate, and dicarboxylic acids (Fig. 7A). At the same time, TA prominently turned on the genes involved in peroxisomal biogenesis, peroxisomal organization, and transport (Fig. 7B).

**(iii) Secondary metabolites.** We also examined the upregulation of secondary metabolite genes in TA and contrasted it against AO. During the metabolite stage, TA displayed high-level expression of seven genes related to the NRPS family, two genes each connected to polyketide synthase (PKS) and PKS-like, and other single genes associated with NRPS-like and NRPS-PKS hybrid families (Fig. 8C). At the same time, AO showed single NRPS and NRPS-like genes and two PKS genes upregulated (Fig. 8A).

Then, in the mycoparasitic stage, TA silenced the NRPS genes and only upregulated PKS and NRPS-like genes and one gene representing dimethylallyl tryptophan synthases (DMATS) (Fig. 8D), while AO also showed a very similar profile with two genes related to DMATS and PKS and one gene associated with NRPS-like genes (Fig. 8B).

**(iv) Quinolinic acid production in *Armillaria*.** GO enrichment results suggested the possibility of significant quinolinic acid production in AO (Fig. 6). To test this possibility at the gene expression level, we aimed to identify the homologous genes in AO involved in the stepwise conversion of tryptophan to quinolinic acid (QA) by applying best-hit reciprocal BLASTs with yeast *BNA* orthologues involved in the biosynthesis of QA (Fig. S4). Orthofinder identified 4,861 orthogroups in AO, among them orthogroups OG0000948 (*BNA2*), OG0002535 (*BNA4*), OG0002534 (*BNA5*), and OG0004139 (*BNA1*) represented indoleamine 2,3-dioxygenase (EC:1.13.11.52) (ARMOST_04226, ARMOST_13362, ARMOST_13365), kynurenine 3-monooxygenase (EC:1.14.13.9) (ARMOST_03616), kynureninase (EC:3.7.1.3)

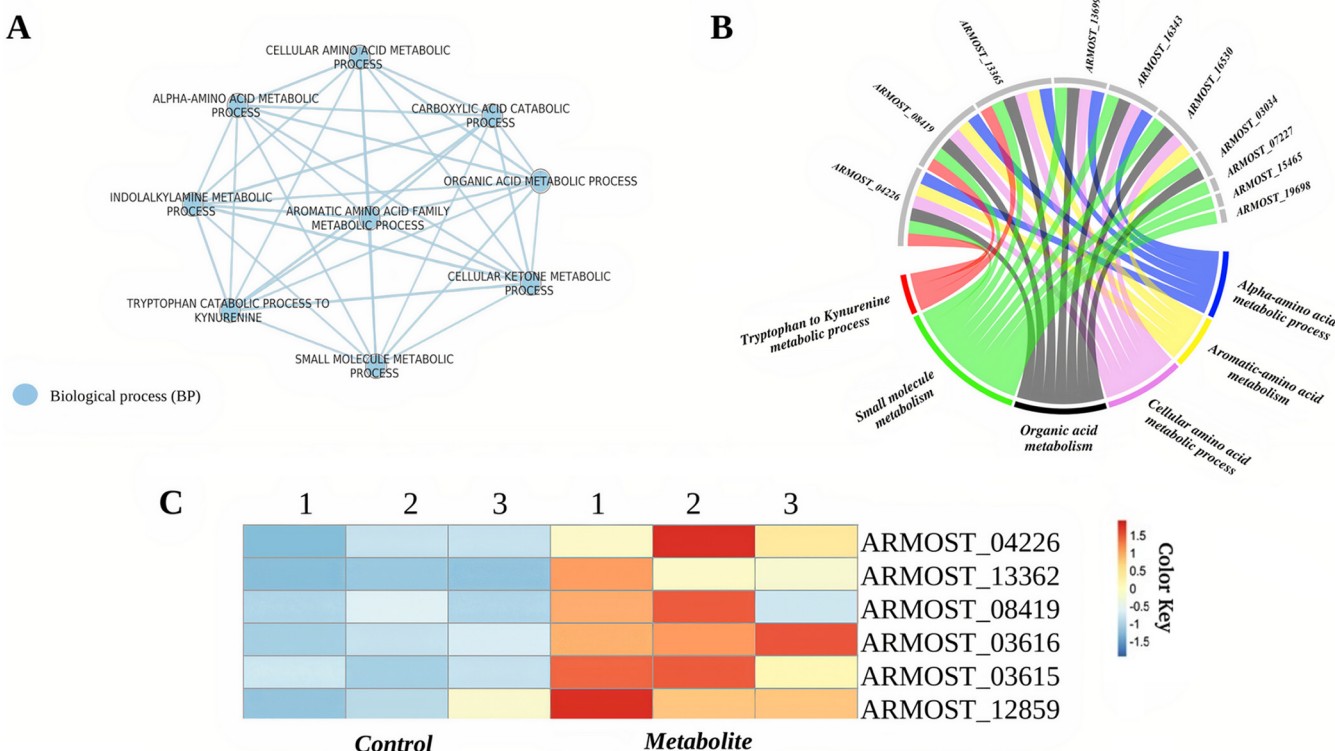

**FIG 6** *A. ostoyae* (AO) metabolite stage GO enrichment. (A) GO enrichment map of the metabolite stage cluster genes from AO. The blue-colored nodes denote the GO type. (B) Chord diagrams of genes contributing to biological processes are shown in the enrichment map. (C) Heatmap plot of genes involved in quinolinic acid (QA) production in *A. ostoyae* at the metabolite stage of interaction (based on yeast homologs, pathway shown in Fig. S3). Genes highly expressed are indicated in red, and lowly expressed are in blue.

(ARMOST_03615), and 3-hydroxyanthranilate 3,4-dioxygenase (EC:1.13.11.6) (ARMOST_12859), respectively. The kynurenine formamidase, involved in the second step of tryptophan degradation, was identified as an *Armillaria*-specific gene (ARMOST_08419), with orthogroup OG0000326, using InterPro analysis (IPR007325). Then their expression profiles were examined, and the results showed that all *BNA* homologs as well as the *Armillaria*-specific kynurenine formamidase gene showed increased expression in AO during the metabolite stage of the interaction (Fig. 6C).

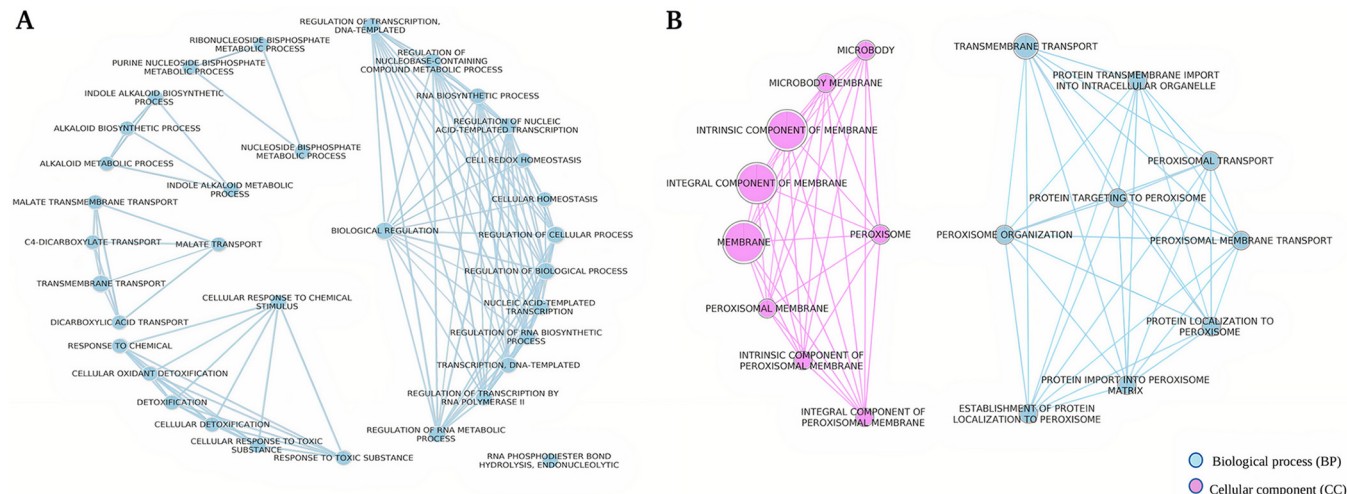

**FIG 7** GO enrichment maps of mycoparasite stage clusters. The mycoparasite stage cluster genes from *A. ostoyae* (AO) (A) and the mycoparasite stage cluster genes from *T. atroviride* (TA) (B). Blue and purple colored nodes represent the type of GO and the edges denote gene overlaps.

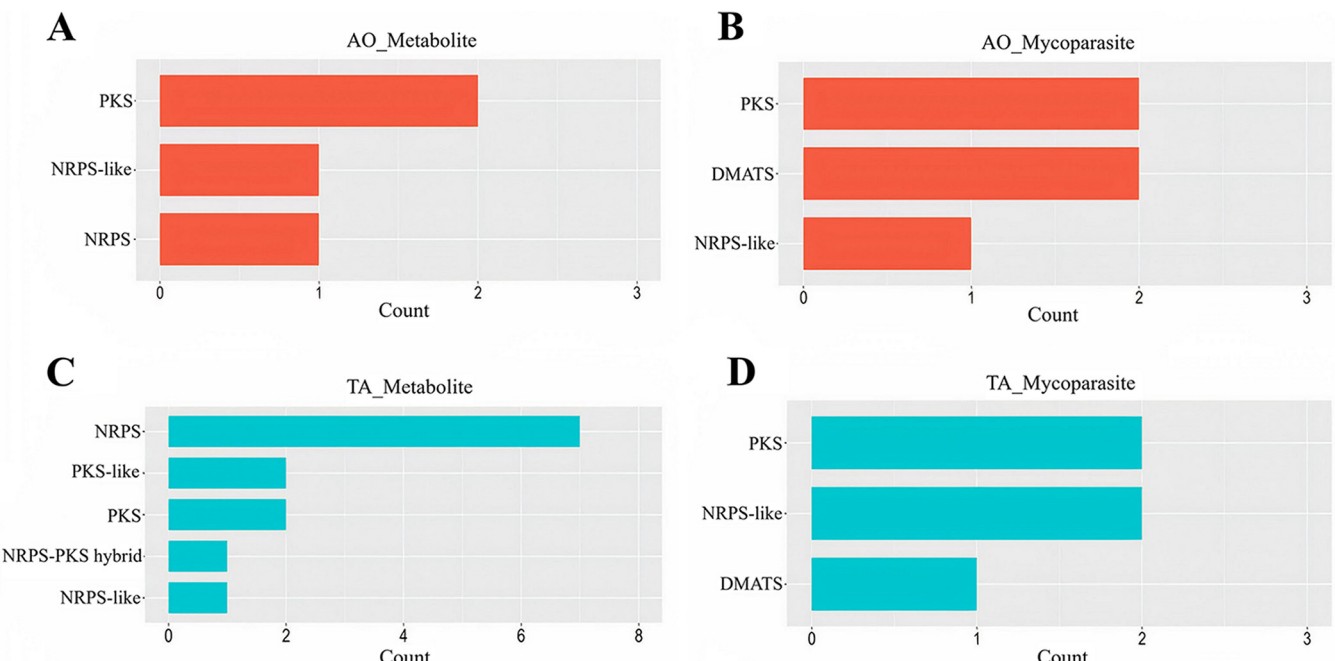

**FIG 8** Secondary metabolite related gene counts in the metabolite and mycoparasite stages. The gene counts are shown in *A. ostoyae* (A and B) and *T. atroviride* (C and D) during the metabolite (A and C) and mycoparasite (B and D) stages of interaction. The *y* axis denotes the metabolite biosynthesis gene families, and *x* axis represents the gene counts.

**Differential gene expression analyses of genes prevalent during the interaction in *A. ostoyae* and *T. atroviride*.** We examined four groups of genes that might be connected to the defense strategies *A. ostoyae* employs against *Trichoderma* (Supplemental File S3 and S4; Fig. 9). Each gene group had differentially expressed genes (DEGs), which could indicate their involvement in the interaction between the two species.

**(i) Defense-related genes in AO.** We used a set of 22 Uniprot IDs of defense genes from studies of bacterial-fungal and nematode-fungal interactions (18–21) and identified their homologs in *A. ostoyae* using MMseqs2 search (E value <1E-20). Out of 22 defense genes, only 6 had hits in *A. ostoyae*, corresponding to 26 proteins. Four of these 26 proteins (Supplemental File S3) were upregulated during both the metabolite and mycoparasite stages of the interaction (highest fold change [FC] = 65.86 in metabolite and 119.58 in mycoparasite stage), and 2 were upregulated only in the mycoparasite stage. These 6 proteins matched the Cop6 alpha cuprenene synthase (Uniport ID A8NCK5) and were identified in *Coprinopsis cinerea* (22) and reported as part of a biosynthetic gene cluster, catalyzing the cyclization of farnesyl/geranyl diphosphate to alpha-cuprenene. Alpha-cuprenenes, which belong to sesquiterpenes, are volatile metabolites produced during fungal development and/or in response to stress. In *C. cinerea*, the *Cop6* gene had the InterPro annotation "trichodiene synthases (IPR024652)," and genes with this term were found to be developmentally regulated in fruiting body formation in *C. cinerea* and *A. ostoyae* (23). This indicates the role of alpha-cuprenene synthases in both development-encoded and induced defense mechanisms.

**(ii) Fungal cell wall degrading CAZymes.** To test the hypothesis that *A. ostoyae* may defend itself by secreting carbohydrate-active enzymes (CAZymes) acting on *Trichoderma*'s cell wall, we examined CAZymes active on fungal cell wall components. We used a set of fungal cell wall degrading (FCWD) CAZy families (Supplemental File S3; reference 24) and examined their expression during the interaction. We identified 110 proteins with FCWD-related CAZyme annotation and found that 6 were upregulated in both the metabolite and mycoparasite stages of interaction, 1 was upregulated only in the metabolite stage, and 11 were upregulated only in the mycoparasite stage (Fig. 10). Among the most upregulated genes, the maximum fold changes ranged from 2.12 to 21.08 times for families such as mannanases (GH76), glucanases (GH128 and

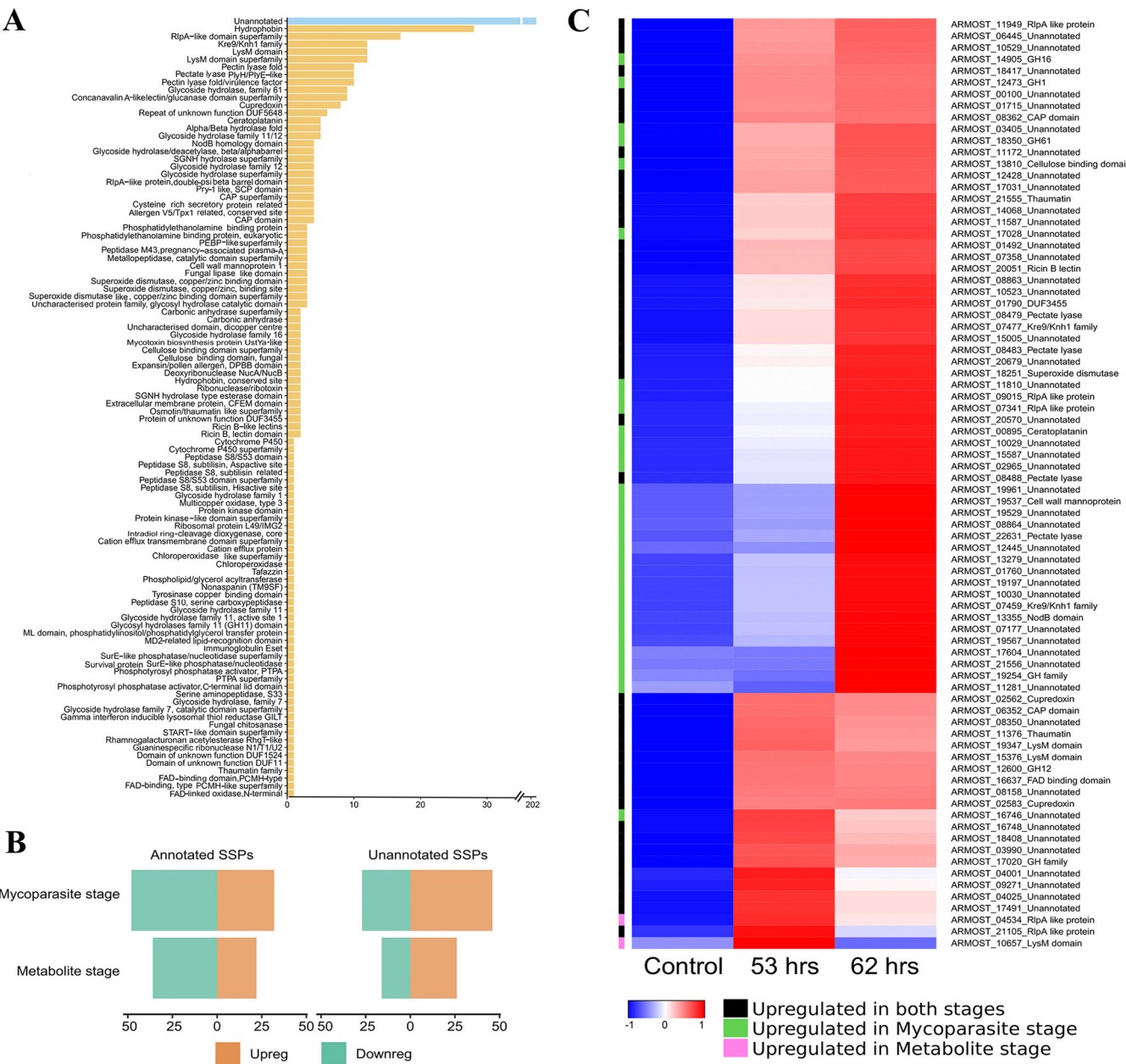

**FIG 9** Small secreted proteins (SSPs) in *A. ostoyae*. (A) Unannotated and InterPro annotated SSPs (*x* axis) and their numbers (*y* axis) in *A. ostoyae*. (B) Number of differentially expressed SSPs in the metabolite stage (53 h) and mycoparasite stage (62 h) of the interaction. Annotated differentially expressed SSPs are shown on the left, and the unannotated ones are on the right. Upregulated and downregulated SSPs are in orange and green, respectively. (C) Heatmap depicting expression of differentially expressed SSPs. The color bar on the left shows at which stage they were upregulated.

GH16), and chitinases/peptidoglycan hydrolases (GH18). Another CAZy family, CBM50, reported to be responsible for chitin and peptidoglycan binding, was found to be upregulated, with the highest fold changes over 308.65 and 212.07 in the metabolite and mycoparasite stage, respectively. From a total of 15 CBM50 genes in *A. ostoyae*, 5 were upregulated in the metabolite stage, and 5 were upregulated in the mycoparasite stage; out of these 4 were commonly upregulated in both stages with a minimum of 24-fold changes. These observations highlight a putative defense strategy the fungus deploys where it tries to defend itself by targeting the cell wall components of the interacting fungus. We detected 40 FCWD-related genes expressed in TA, among them 33 upregulated during the 0-h or the metabolite stage of the interaction (Fig. 10). Of the FCWD-related genes, 14 encoded chitin-binding proteins (12 GH18, 1 GH20, and 1

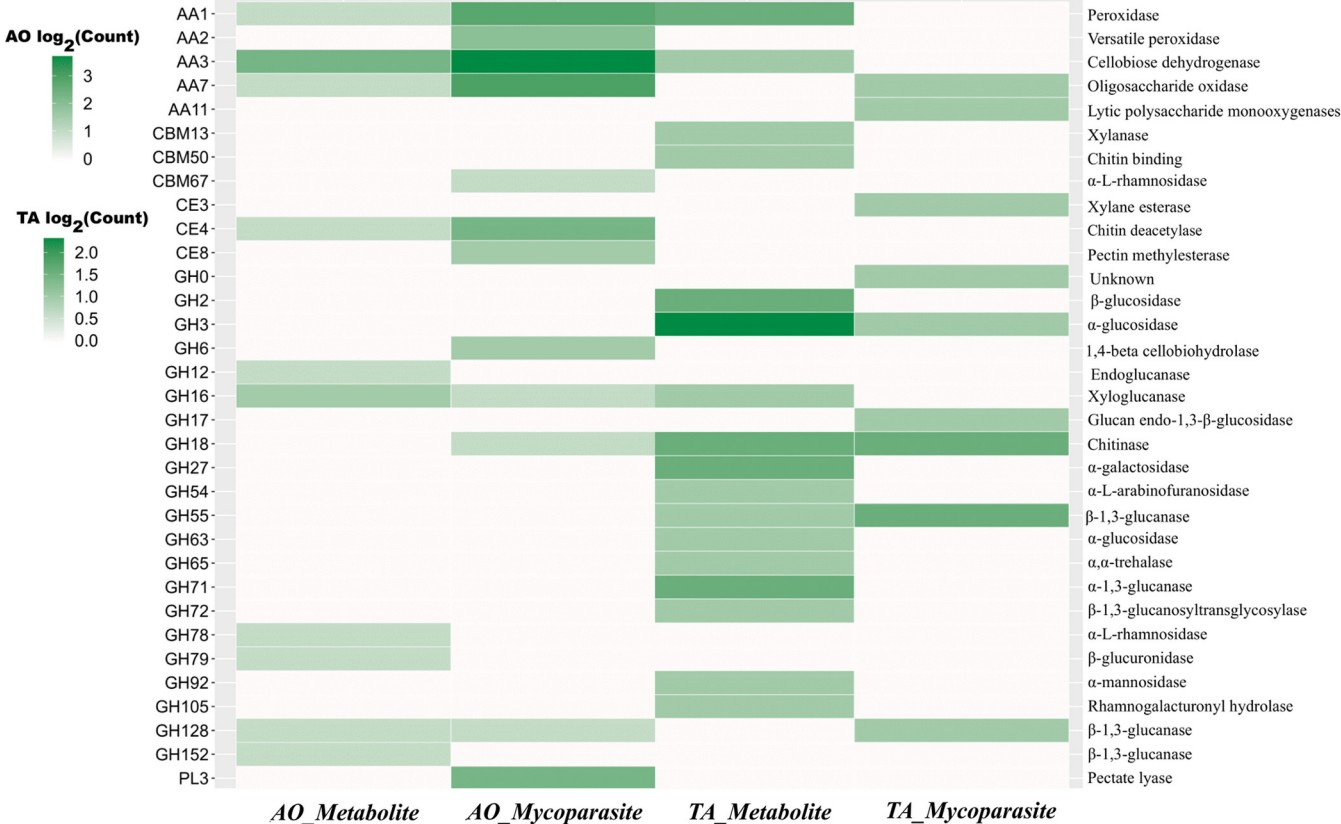

**FIG 10** Differentially expressed CAZymes in AO and TA. Heatmap plot depicting the counts of diffentially expressed CAZymes in both AO and TA at the metabolite and mycoparasite stages. The *x* axis denotes the organism and stage, whereas the *y* axis on the left side indicates the CAZyme family and the InterPro families on the right.

GH7), 10 mannanases (6 GH92, 3 GH76, and 1 GH125), 8 glucanases (4 GH55, 3 GH71, and 1 GH128) and 1 was unclassified (GH30_5). GH72, which is also a glucanase, was uniquely upregulated only in the metabolite stage. We identified only 6 FCWD-related genes expressed during the mycoparasitic stage. Among the 6 genes, 5 belonged to the chitinase/peptidoglycan hydrolase (GH18) family, and 1 represented the endo-$\beta$-1,3-glucanase (GH81) family.

**(iii) Small secreted proteins.** We identified 414 small secreted proteins (SSPs) in *A. ostoyae*, of which 382 were expressed in the interaction study (Supplemental File S3; Fig. 9). Of these 382 proteins, 202 had no known InterPro domains, while the remaining 180 proteins corresponded to about 100 unique InterPro terms, including hydrophobins, thaumatins, cysteine-rich CAP domain, LysM (=CBM50) domain, cerato-platanins/RlpA-like domain, ricin B lectin domain-containing proteins, etc. (see Fig. 9A for InterPro terms and their counts). There were 100 differentially expressed SSPs (Fig. 9C) in the control versus metabolite stage (48 upregulated and 52 downregulated in the metabolite stage) and 153 differentially expressed SSPs in the control versus mycoparasite stage (78 upregulated and 75 downregulated in the mycoparasite stage). There were 46 upregulated and 48 downregulated genes in both the metabolite and mycoparasite stages. In both comparisons, the upregulated genes had higher proportions of unannotated SSPs than annotated ones (Fig. 9B). The Fisher-exact test revealed that SSPs were significantly overrepresented among DEGs (Supplemental File S3) when upregulated, and downregulated genes were both considered and when only upregulated SSPs were considered. This suggests that SSPs are a particularly important group in the interaction; however, ascertaining how they contribute mechanistically is difficult because they are mostly unannotated proteins. Studies on fungal-fungal (25) and fungal-plant interactions (26–28) have speculated about the possibility of SSPs as putative effectors but have rarely reported

anything concrete concerning specific functional roles. When comparing the expression of SSPs in TA at the metabolite stage and the mycoparasitic stage, we identified 157 differentially expressed SSP-related genes (Supplemental File S4). Of the DEGs, 43 showed upregulation in the control and the metabolite stage of interaction. Among those DEGs, 25 had no Interpro annotations, while the remaining 18 corresponded to Interpro terms like cerato-platanin, cerato-ulmin hydrophobin, RlpA-like protein, cutinase, cysteine-rich secretory protein related with CAP domain, RNase T2-like, phytocyanin domain, peptidase S51, blastomyces yeast-phase-specific protein, CFEM domain-containing proteins, DUF1349, fungal chitosanase, cyanovirin-N, alginate lyase 2, Kre9/Knh1 family, cell wall mannoprotein 1, and uncharacterized glycosyl hydrolase. Altogether, 13 SSPs were upregulated during the metabolite stage. Of these, 4 were cerato-ulmin hydrophobins, 1 was CFEM, and 1 was chloroperoxidase; the rest were unknown. In addition, there were 8 SSPs that were upregulated in both the metabolite and mycoparasite stage, and 4 of them were cerato-ulmin hydrophobin, cerato-platanin, DUF1524, and cell wall mannoprotein 1. During the mycoparasite stage in TA, 3 G1 peptidase genes were upregulated along with other genes like *Alternaria alternata* allergen 1, cerato-ulmin hydrophobin, CFEM, FAS1 domain-containing protein Mug57-like, guanine-specific RNase N1/T1/U2, and serine proteases with trypsin domain.

**(iv) Peptidases.** In host-pathogen interaction, secreted peptidases might act as virulence factors by altering the protein components of the host (29). Therefore, we identified peptidases in AO (Supplemental File S3) and TA (Supplemental File S4) using the MEROPS database. We identified the upregulation of 151 differentially expressed peptidase-related genes in AO of which 39 were upregulated during the mycoparasite stage, and 37 peptidase-related genes in both the metabolite and mycoparasite stages (Fig. 11). Aspartic peptidase A1A and serine peptidase S12 displayed the highest expression in the mycoparasite stage, while serine peptidases (S09, S33) and cysteine peptidase C12 were dominant in the metabolite and mycoparasite stages of the interaction. In TA, there were 52 differentially expressed secreted peptidases (Fig. 11). Among them, 17 peptidases upregulated in the mycoparasite stage of interaction included 4 aspartic peptidases (A01A), 3 glutamic peptidases (G01), and 5 serine peptidases (S01A, S08A, S09X, S12, and S54). One cysteine peptidase C56 was upregulated in both the metabolite and mycoparasite stages. The upregulation of A1 and G1 peptidases during the mycoparasite stage may contribute to the pathogenicity and the host adaptation of TA (29) toward AO.

**(v) Apoptosis-related genes.** During the mycoparasite stage, AO-specific upregulation occurred in 10 apoptosis-related genes (Fig. 12). The ARMOST_05616 and ARMOST_18631 genes belong to the metacaspase 1 (MCA1) family, while ARMOST_18535, ARMOST_18537, ARMOST_09700, and ARMOST_09409 encode apoptosis-inducing factor 2 proteins (AIFM2). Metacaspase genes and AIFM2 may have direct roles in apoptosis whereas bax inhibitors (BI) (30–33), Rho-Rab GTPases, and thioredoxin reductases may be directly or indirectly involved in apoptosis (34–36).

**Validation of differentially expressed genes using quantitative real-time reverse transcription-PCR.** To confirm the reliability of the RNA-Seq data, the transcriptional level of 11 unigenes was examined by quantitative real-time reverse transcription-PCR (qRT-PCR) (Fig. 13), including ARMOST_13362, ARMOST_03616, ARMOST_5857, ARMOST_18537, ARMOST_05856, ARMOST_05616, ARMOST_04226, and ARMOST_18535 for *A. ostoyae*, as well as XM_014093434.1, XM_014092940.1, and XM_014085775 for *T. atroviride*. In response to the interaction, all eight *Armillaria* genes exhibited higher expression at the metabolite and mycoparasite stages (before physical contact and during physical contact) than the control stage (Fig. 13A). The qRT-PCR analysis confirmed the upregulation of the ARMOST_04226, ARMOST_13362, and ARMOST_03616 genes, involved in the first biosynthetic steps of quinolinic acid, in the metabolite stage (Fig. 6C). Except for two apoptotic genes (ARMOST_05616 and ARMOST_18535), the *Armillaria* genes exhibited no significant differences between the metabolite and mycoparasite stages, suggesting that the expression of the tested genes was activated before physical contact and did not change during the physical interactions. In contrast, the *Trichoderma* genes showed

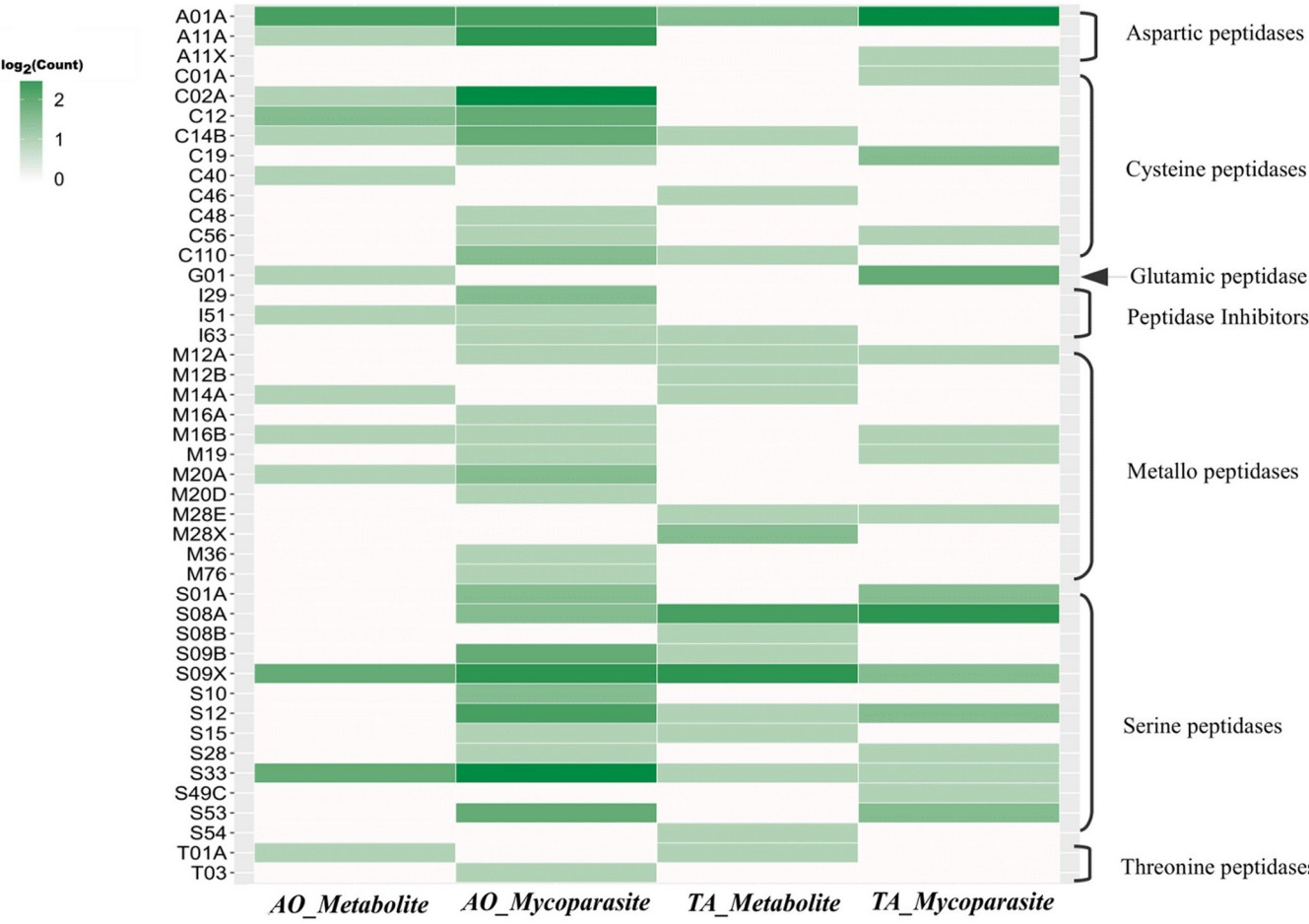

**FIG 11** Differentially expressed proteases in AO and TA. Heatmap plot of differentially expressed protease counts in both AO and TA. The x axis represents the organism and stage; the y axis on the left side denotes the peptidase family and on the right side the major peptidase groups.

continuously increasing expression levels during the interaction (Fig. 13B) and became most active in the mycoparasitic stage. Altogether, these unigenes were upregulated in comparison with the control, consistent with the RNA-Seq data, indicating that our experimental results were valid.

## DISCUSSION

As little has been known about their intricate molecular interplay during the mycoparasitic process, we used dual culture assays to examine the interaction between *Trichoderma* and the target pathogen *Armillaria*. We have selected a haploid AO isolate for the tests as haploids form homogenous, evenly growing colonies compared to the variable-shaped AO diploids and react with easily noticeable sensitivity to the approaching invasive *Trichoderma* mycelia. In our experiments, AO sensed and responded to the presence of the neighboring invader TA and altered its behavior accordingly. Yet, TA mycelia could grow onto and inhibit the growth of the haploid AO colonies (Fig. 1).

In this study, complex responses were observed in AO, reflected in a continuous downward trend in gene expression profiles from 53 h (before physical contact) to 62 h (during physical contact). The genes showing a downtrend included those related to cell cycle control, as revealed by the enrichment of genes involved in DNA replication, DNA repair, mitotic cell cycle, and microtubule-based processes. The downtrend of gene clusters related to the cell cycle indicated AO growth regression when the mycelia of TA gradually approached and then dominated the AO colony.

By that time, AO had implemented several self-protecting mechanisms against TA. First, at the transcriptional level, AO responded to the approaching TA, which was

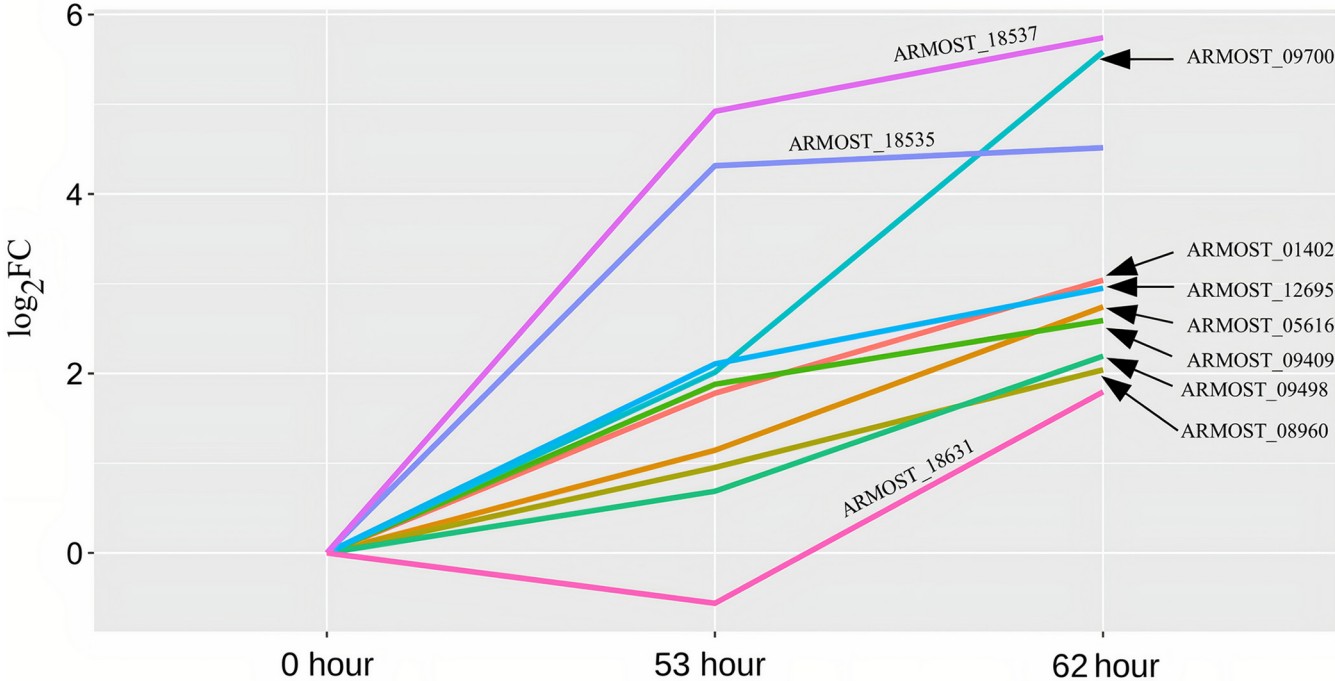

**FIG 12** Apoptosis-related gene expression in *A. ostoyae*. Line plot of apoptosis-related gene expression profiles in *A. ostoyae* during the mycoparasite stage. The *x* axis denotes the time points, and the *y* axis indicates the $log_2$-fold change. The lines are marked with the geneID it represents.

revealed by reactions that identified TA as an antagonistic intruder. These reactions included countering oxidative stress, activating defense processes, and metabolizing toxic compounds.

Gene activations related to oxidative stress relief, including the glutathione peroxidase gene before the physical contact and others with DSBA-like thioredoxin domains and NADH:flavin oxidoreductase/NADH oxidase (NOX) genes during physical contact, were detected in AO. The higher expression levels of a glutathione peroxidase gene might have been critical for the resistance to oxidative stress and then the survival of AO cells (37, 38). The potential role of NOX genes in AO is unclear; they may participate in a process that induces a defensive system by increasing the intracellular $H_2O_2$ levels (39); however, the latter may also trigger a rapidly progressing apoptotic self-destruction process (40).

Defense-related functions included the upregulation of genes involved in the biosynthesis of antibiotics and in the neutralization of toxic compounds. Genes with condensation and SnoaL-like domains implicated in the biosynthesis of polyketides (PKs) and nonribosomal peptides (NRPs) were dominant in AO before the physical contact (Fig. 8A). SnoaL belongs to a family of small polyketide cyclases responsible for the biosynthesis of polyketide antibiotics (41). Regarding the control of toxic metabolites, a phenol hydroxylase involved in the hydroxylation of phenolic compounds (42) and flavin-dependent halogenases (Fl-Hals) acting on aromatic substances (43) were also found in AO (Fig. 5A). Hydroxylation of phenolics and introducing a halide component to aromatics, thereby derivatizing and changing bioactivity and metabolic susceptibility of potentially toxic compounds, may have also contributed to the self-defensive activities of AO.

At the metabolite stage, indoleamine 2,3-dioxygenase (IDO) was highly expressed in AO (Fig. 5A). IDO is a tryptophan-degrading enzyme supplying NAD ($NAD^+$) via the kynurenine pathway in fungi (44, 45). Correspondingly, upregulation of the kynurenine pathway in AO probably leads to the production of an intermediate, the quinolinic acid (QA), at the metabolite stage (Fig. 6A). Ohasi et al. (46) first reported the secretion of QA into the medium by *S. cerevisiae* and the utilization of the secreted QA as the

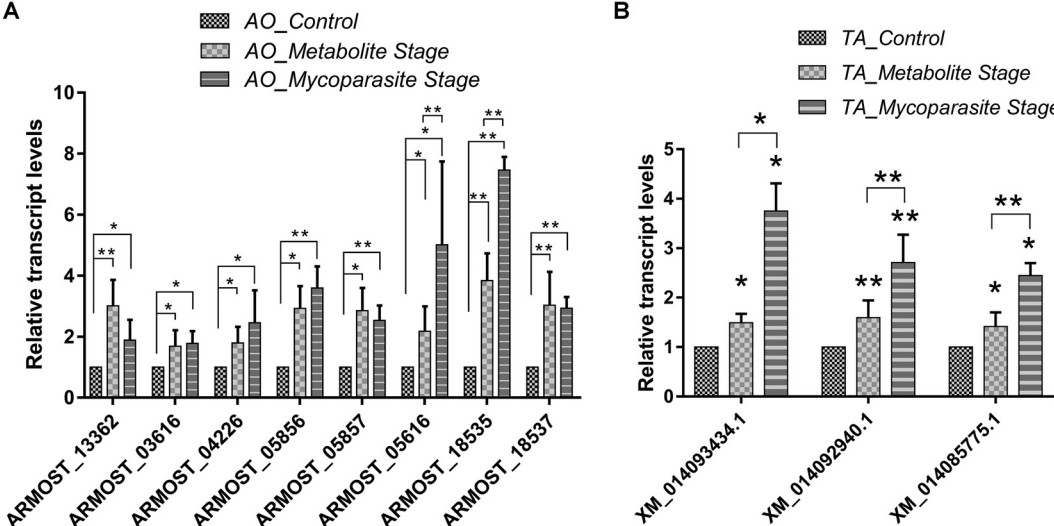

**FIG 13** qRT-PCR validation of differentially expressed genes in *A. ostoyae* and *T. atroviride*. The figure represents the qRT-PCR validation of the selected 8 and 3 genes that were differentially expressed at the metabolite and mycoparasite stages of the interaction in *A. ostoyae* (A) and *T. atroviride* (B), respectively. The actin and the *gpd* genes were the internal controls for validating the *Armillaria* and the *elF-4* gene for the *Trichoderma* genes. The presented values are averages of three independent experiments; error bars indicate standard deviation. Asterisks represent statistical significance (**, $P < 0.01$ and *, $P < 0.05$).

precursor of $NAD^+$. QAs are formed in the media from secreted precursors (Fig. S4) for storage and later use as QA can be quickly absorbed and used to maintain intracellular $NAD^+$ concentration. Therefore, it is possible that AO stores abundant QA extracellularly which is then transformed into $NAD^+$ to maintain cellular viability and to cope with the high stress induced by TA. Notably, the antifungal properties of QA, such as inhibition of fungal mycelia, fungal cell wall alterations, and disease incidence reduction, were studied in *Ceratocystis fimbriata*, a fungal pathogen causing black rot disease on sweet potato (47). Therein, quinolinic acid was demonstrated to have antifungal activity by interfering with the ergosterol biosynthesis pathway. During the metabolite stage of the interaction, ergosterol biosynthesis was enriched in TA (Fig. S3); hence, it could be speculated that quinolinic acid might have triggered such a response.

Additional genes were selectively upregulated during the mycoparasitic stage, which may also protect AO against the invading fungus. These genes included malic acid transporters possibly for an increased extracellular release of malic acid (Fig. 5C). Malic acid has been shown to have antimicrobial and antifungal activities against various bacterial and fungal species and has been used as an antimicrobial agent against *Listeria monocytogenes*, *Salmonella enterica*, *Escherichia coli*, and *Rhizopus nigricans* (48, 49). Notably, malic acid was found to be the most abundant organic acid in *A. mellea* fruiting bodies collected from nature (50), which may represent a natural antimicrobial defense option for *Armillaria* species.

In the case of TA, major transcriptional differences we identified were associated with oxidative stress tolerance and complex metabolic pathways in the presence of AO, highlighted by cytochrome $b_5$-like heme-binding domain-containing proteins and numerous genes encoding SDRs (Fig. 5D) (51, 52).

NRPSs play an important role during the interaction of biocontrol fungi with plant pathogens, insects, and plants (53). We predicted the genes involved in secondary metabolite biosynthesis (e.g., NRPS, PKS-like, PKS, NRPS-PKS hybrid, and NRPS-like), and these were highly expressed in TA at the metabolite stage (Fig. 8C), suggesting that TA actively antagonized AO through the production of antimicrobial compounds. This was indicated particularly by the significant expression of NRPS genes already at the metabolite stage. These expression profiles signified that TA activated antagonistic processes before physical contact with AO and that AO reacted by activating detoxification mechanisms and defense

processes. On the other hand, the expression of genes encoding toxic secondary metabolites was underrepresented in AO at the metabolite stage compared to TA. In addition, fungal ABC transporters (54) functioning in cellular detoxification were also highly expressed in TA during the mycoparasite stage (Fig. 5D).

Overall, TA highly expressed ABC transporters, as well as genes producing toxic secondary metabolites (55, 56). ABC transporters possibly involved in releasing antifungal and other toxic compounds became prevalent in TA when its mycelia gradually reached and physically contacted AO.

At the mycoparasite stage, TA activated peroxisome-related processes (Fig. 5D and 7B). Peroxisomes are involved in lipid metabolism and associated with several essential metabolic pathways and the homeostasis of reactive oxygen species (57). The peroxisome-related activities also represent a type of defense system that aims to aid the survival of the multicellular organism (58). Peroxisomes may play an essential role in *Trichoderma* survival when approaching prey, coping with prey-derived signals and chemicals, and developing contact-induced morphogenesis at physical contact with AO (Fig. 7B). At the end, apoptosis-like cell death seems to occur in AO, reflected by the significant upregulation of apoptosis-related genes once TA is physically in contact with AO (Fig. 12). Apoptosis can be induced in fungi via exposure to toxic metabolites or other stress factors (33).

In *Trichoderma* species, CAZymes are responsible for the adaptive remodeling of their cell wall and the degradation of the cell wall residues of their fungal prey during mycoparasitic activities (59). According to Schmoll et al. (60), the different ecological behavior of the mycoparasites, such as *T. virens*, *T. atrobrunneum*, and TA, compared to the industrially important cellulolytic *T. reesei* (with a genome size of 34.1 Mbp), is reflected by the expansion of their genome size (59). A higher number of CAZyme domains was found in the expanded genomes of the mycoparasites, such as TA IMI206040 (36.1 Mbp) and *T. atrobrunneum* ITEM 908 (39.2 Mbp) (60). The present study performed a focused investigation on the CAZyme dynamics in the transcriptomes of AO and TA during the metabolite and mycoparasite stages of their interaction (Supplemental File S3 and S4). In the transcriptomes of TA affected by AO, we detected differential expression of CAZymes at the metabolite and mycoparasite stages, including auxiliary activities (AAs) of redox enzymes that act with CAZymes, carbohydrate esterases (CEs) responsible for the hydrolysis of carbohydrate esters, and glycoside hydrolases (GHs) that associate with the hydrolysis and/or rearrangement of glycosidic bonds (www.cazy.org). Most FCWD enzymes were strongly expressed in AO in the metabolite and mycoparasitic stages, while in TA, they were already significantly present in the precontact phase. The latter finding agrees with Kullnig et al. (61), where certain enzymes, such as chitinases, were already secreted in the precontact phase.

Among the different GH families of TA, GH18 represents chitinases, the upregulation of which can increase the ability of TA to colonize its prey cells (62–64). The activation of genes encoding chitinolytic enzymes and glucanases is likely to play a significant role in the mycoparasitism of *Armillaria*, which may be synergistic with the potential antimicrobial effect of various secondary metabolites (8). As the incubation time extended, the expression of GHs in TA decreased significantly during the mycoparasite stage, presumably due to a possible saturation of these enzymes in the media. In contrast to TA, lower abundance and diversity of CAZymes were found in AO before its physical contact with TA. Later, these CAZymes showed no significant change, except for several AAs of the redox enzymes that were significantly upregulated at the mycoparasite stage, which may also be part or consequence of a defensive response against oxidative stress.

Since glycoproteins are vital structural components of the fungal cell walls, the participation of secreted peptidases and proteases in the prey-targeted cell wall degradation seems unmissable in *Trichoderma* mycoparasitism (65). Similarly to CAZymes, many of the secretory protease genes were already expressed during the precontact stage in TA and relatively less in AO (Fig. 11; Supplemental File S3 and S4). However, in

the mycoparasite stage, the abundance and diversity of the protease profiles increased extensively in AO, while, with comparable abundance levels between the metabolite and mycoparasitic stages, the specificity and variability of protease activities changed significantly in TA. All changes at the mycoparasite stage seem to be induced by the direct contact-level mycoparasite-prey interaction.

The possible contribution of proteases to the degradation of the fungal cell wall during *Trichoderma* mycoparasitism and their putative involvement in the interactions with different organisms triggered great investigative interests. For example, the gene *papA* encoding one of the extracellular aspartyl proteases from *T. asperellum* was upregulated during plate confrontation with *R. solani* (66). Moreover, peptidases also act as proteolytic inactivators of virulence-related enzymes or other pathogenicity factors from pathogens; certain metalloendopeptidases, serine proteases, and aspartic proteases were induced in *T. harzianum* during *in vitro* nematode egg-parasitism of *Caenorhabditis elegans* and revealed a substantial biocontrol role of these proteases in this function (67). On the other hand, the functions of peptidases and proteases in the detoxification of toxic molecules have been confirmed in *Aspergillus niger* and *R. solani* in response to biocontrol bacteria from the genera *Serratia* and *Bacillus* (68, 69). The detoxifying function of peptidases appears to be fully activated and implemented in AO. As the mycelia of TA extended toward interacting with AO, AO showed an intense reaction with significantly more types and increased peptidase production (Fig. 11), suggesting that TA activates a typical defense process in AO.

## MATERIALS AND METHODS

**Strains and culture conditions.** *Trichoderma atroviride* SZMC 24276 (TA) was isolated from a soil sample collected in a native spruce forest in Rosalia, Austria (GPS-N: 47°41.640, GPS-E: 16°17.937). The diploid strains of *A. ostoyae*, SZMC 24127, SZMC 24128, and SZMC 24129, were isolated from *Armillaria* fruiting bodies or rhizomorphs obtained previously from the Rosalia forest (GPS-N: ranging from 47°41.618 to 47°41.629, GPS-E: 16°17.873 to 16°17.964) (16). The diploid strain C18 of *A. ostoyae* is a field isolate from Switzerland (70), while the haploid strains of *A. ostoyae*, C18/9, C18/2, C18/3, and C18/4, were derived from C18 as single spore isolates. All *Trichoderma* and *Armillaria* strains were deposited in the Szeged Microbiology Collection (SZMC, www.szmc.hu) in Szeged, Hungary. Corresponding SZMC numbers of the five *A. ostoyae* strains C18, C18/9, C18/2, C18/3, and C18/4 are SZMC 23083, SZMC 23085, SZMC 27047, SZMC 27048, and SZMC 27049, respectively. All fungal strains were cultured on PDA medium (VWR, Debrecen, Hungary).

***In vitro* antagonistic activity assessment by dual culture assay.** The *T. atroviride* SZMC 24276 strain and the haploid strain *A. ostoyae* SZMC 23085 were screened and determined for further mRNA sequencing experiments based on their confrontation interactions. Using dual-culture confrontation tests, we examined *T. atroviride* SZMC 24276 *in vitro* for its antagonistic ability against both diploid strains and haploid derivatives of *A. ostoyae*. In the experiments, agar plugs (5 mm in diameter) of *A. ostoyae* cut from a PDA plate of a 4-week-old colony were inoculated on a PDA medium about 2 cm near the edge of the Petri plates. After 3 weeks of incubation at 26°C, agar plugs (5 mm in diameter) of *T. atroviride* excised from a PDA plate of a 2-day-old colony were inoculated 3.5 cm from the edge of the *Armillaria* colonies, resulting in a confrontation growth of the two toward each other. We took plate photographs after another 5 days of coincubation at 26°C. Image analysis of plate photographs was performed using the ImageJ software, and then biocontrol index (BCI) values were calculated based on the following formula (71):

$$\mathrm{BCI} \; = \; \left(\text{area of } \textit{Trichoderma} \text{ colony/total area occupied by the colonies of both } \textit{Trichoderma} \text{ and the plant pathogenic fungus}\right)$$
$$\times \; 100$$

All dual culture confrontation experiments were repeated three times under the same test conditions. Values were recorded as the means with standard deviations from triplicates.

**Transcriptome analysis of *Trichoderma atroviride–Armillaria ostoyae* dual cultures. (i) Experimental design, sample collection, and total RNA extraction.** For the time-course analysis of the *Armillaria-Trichoderma* interaction, PDA plates were first inoculated with the haploid *A. ostoyae* strain SZMC 23085 (C18/9) (72) and grown for 21 days at 26°C. Then, on day 22, plates in parallel were coinoculated with *T. atroviride* strain SZMC 24276 pregrown on PDA plates for 2 days. We performed the inoculations using agar plugs (5 mm in diameter) with fungal mycelia. *Armillaria* colonies were inoculated to a position 2 cm near the edge of the Petri plates. After 21 days, the *Trichoderma* colonies were inoculated 3.5 cm from the edge of the *Armillaria* colonies, and the coinoculated plates were incubated further at 26°C. The interactive fungal mycelia from both sides were harvested 53, 62, and 105 h after *Trichoderma* inoculation, representing nonphysical (metabolite stage), physical (mycoparasite stage), and postmycoparasitic (or postnecrotrophic) interaction stages, respectively. While *Trichoderma* and *Armillaria* mycelia were separately harvested at the metabolite stage (53 h), the physically interacting mycelia were

coscraped from the mycoparasitic (62 h) and postmycoparasitic settings (105 h) (Fig. S5). Three biological replicates were considered for each time point, including individually growing, noninteractive cultures (the 21-day-old *Armillaria* and 2-day-old *Trichoderma* colonies) as the controls. All collected mycelial samples were immediately frozen in liquid nitrogen and stored at −80°C.

Four conditions were established and analyzed: 0-h/control samples of *Armillaria* and *Trichoderma* grown separately on PDA media; 53rd hour/metabolite stage samples considering the impact of various metabolites between *Trichoderma* and *Armillaria* before physical contact; 62nd hour/mycoparasite stage samples during the mycoparasitic interaction once *Armillaria* and *Trichoderma* started to contact; and 5th day/postinteractive stage when the *Armillaria* colony was entirely covered with mycelia and conidia of *Trichoderma*. Figure S5 provides details about the culturing of the fungi for the collection of mycelia.

Total RNA extraction from the mycelial samples was conducted with the E.Z.N.A. Plant RNA kit (Omega Bio-Tek Inc. Norcross, GA, USA) according to the manufacturer's extraction protocol with minor modifications. Briefly, mycelia were transferred into an autoclaved mortar, frozen under liquid nitrogen, and immediately ground with an autoclaved pestle before the samples thawed. Degrading RNA content was first estimated using agarose gel (1%) electrophoresis. RNA concentration and qualification were monitored using the Tapestation 2200 analyzer (Agilent Technologies, Santa Clara, CA, USA).

**(ii) cDNA library preparation, sequencing, and data analysis.** We prepared cDNA sequencing libraries for the transcriptome samples using the TruSeq RNA Library Prep kit v2 (Illumina, San Diego, CA, USA). Paired-end fragment reads were generated on an Illumina NextSeq sequencer using TG NextSeq 500/550 High Output kit v2 (300 cycles). Primary data analysis (base-calling) was performed with the "bcl2fastq" software (v2.17.1.14 conversion software, Illumina, San Diego, CA, USA). The quality of the raw reads obtained was analyzed using FastQC (73). Further, low-quality bases (Q score <20) were trimmed using Trimmomatic v0.39 (74). We used Salmon v1.1.0 (75) to quantify the transcripts and generate a count matrix.

**(iii) Time-course analysis.** The TCSeq R-package (76) was utilized to perform c-means clustering analysis of time-course RNA-seq data based on the gene expression profiles. The gene clustering was performed using the Euclidean distance method and the Fuzzy C-means clustering algorithm. Z-scores of gene expression values were used to generate a cluster plot for visualization. To plot the temporal expression values of the genes that were grouped into clusters (Fig. 3), the "timeclustplot" function of TCSeq was used.

Regarding the differential expression analysis, TCSeq was employed to detect differentially expressed genes between different time points. TCSeq uses the generalized linear model and statistical tests such as the likelihood ratio tests of the edgeR package (77) to determine the genes differentially expressed between time points. $Log_2$ FC values of gene expression levels were calculated for pairwise comparisons, and statistically significant differentially expressed genes were filtered using a threshold of $P$ value < 0.05 and $log_2$ FC > |1|.

From the clusters (Fig. 3), we manually filtered and grouped genes according to their expression trends (upregulation/downregulation) (Table 2) at different stages. Downtrend genes included gene sets from Cluster 1 that specifically showed continuous downregulation in either AO or TA. In contrast, the metabolite and mycoparasite cluster genes (Table 2) are those genes from cluster 2 and cluster 3 (Fig. 4) that demonstrated the highest upregulation in the metabolite stage (53 h) or the mycoparasite stage (62 h), respectively. The criteria chosen for this analysis were a minimum fold change >2 and a $P$ value <0.05 between any two time points in a cluster.

Regarding Fig. 9, the scales were generated using the z-scores, calculated by the heatmap.2 function from the gplots package in R. For the input file, the averaged expression values were used.

**(iv) Functional annotation of genes.** Amino acid sequences from *A. ostoyae* (accession number: GCA_900157425.1) (72) and *T. atroviride* (accession number: GCF_000171015.1) (59) were used for secretory protein prediction.

We used the PANNZER2 (Protein ANNotation with Z-scoRE), a fast functional annotation web server (78) for the Gene Ontology (GO) annotation, and InterProScan v5.38 (79) for the functional characterization of proteins. We performed CAZy annotation using the dbCAN2 pipeline (80) and identified proteases using the Diamond BLAST against the MEROPS database (1E-10) (81). The NetGPI (82) online server was used for glycosylphosphatidylinositol (GPI) anchor prediction. Secondary metabolites-related gene/protein prediction was performed using the Secondary Metabolites with the InterProScan (SMIPS) prediction tool (83). We performed Kyoto Encyclopedia of Genes and Genomes (KEGG) annotation using KofamKOALA scoring criteria (84). Transporter proteins were predicted using the Transporter Classification Database (TCDB) (85). GO enrichment analysis was performed in Cytoscape v3.7.2. (86) using BiNGO v3.0.3 (87) and enrichmentMap v3.3.1 (88) plugins (adjusted $P$ value <0.05). The Clusterprofiler R-package (89) was used for InterPro enrichment analysis. We generated all images using the ggplot2 (90) R package.

**(v) Identification of the AO genes involved in the biosynthesis of quinolinic acid.** We identified the AO genes homologous to *Saccharomyces cerevisiae* genes involved in the quinolinic acid biosynthesis using OrthoFinder (91), which relies on RBNHs (Reciprocal Best length-Normalized Hits) to delimit orthogroup sequences based on sequence similarities before MCL (Markov Cluster Algorithm) after all_vs_all BLAST.

**(vi) Quantitative real-time reverse transcription-PCR.** For qRT-PCR analysis, total RNA samples were extracted using the E.Z.N.A. Plant RNA kit (Omega Bio-Tek Inc. Norcross, GA, USA) according to the manufacturer's extraction protocol. The quality of each RNA sample was checked in 2% agarose gel. cDNA synthesis was performed by using Maxima H Minus First Strand cDNA Synthesis kit (ThermoFisher Scientific, Waltham, MA, USA). Oligo (dT)18 and random hexamer primers were used in the reaction mixture according to the manufacturer's instructions.

The qRT-PCR experiments were performed in a CFX96 real-time PCR detection system (Bio-Rad, Hercules, CA, USA) using the Maxima SYBR green qPCR Master Mix (Thermo Scientific, Waltham, MA, USA), and the primers are presented in Table S1. The reaction was realized using the following conditions: denaturation at 95°C for 3 min, followed by 40 cycles of amplification (95°C for 5 s, 60°C for 30 s,

and 72°C for 30 s). The relative quantification of gene expression was executed with the $2^{-\Delta\Delta Ct}$ method (92) using the housekeeping gene glyceraldehyde-3-phosphate dehydrogenase (*gpd*, ARMOST_14637) or actin (ARMOST_03733) for *A. ostoyae* and the *gpd* (ID 297741) or *elF-4* (ID 301614) genes for *T. atroviride* (93). For each sample, 2 technical replicates of the qRT-PCR assay were used with a minimum of 3 biological replicates. Significance was calculated with paired *t* test using the GraphPad Prism 7.00 program (GraphPad Software, La Jolla, CA, USA). $P < 0.05$ was considered statistically significant.

## SUPPLEMENTAL MATERIAL

Supplemental material is available online only.

**SUPPLEMENTAL FILE 1**, XLS file, 0.2 MB.
**SUPPLEMENTAL FILE 2**, XLS file, 1.7 MB.
**SUPPLEMENTAL FILE 3**, XLS file, 0.3 MB.
**SUPPLEMENTAL FILE 4**, XLS file, 0.5 MB.
**SUPPLEMENTAL FILE 5**, XLSX file, 0.01 MB.
**SUPPLEMENTAL FILE 6**, TIFF file, 0.04 MB.
**SUPPLEMENTAL FILE 7**, TIFF file, 1.1 MB.
**SUPPLEMENTAL FILE 8**, TIFF file, 0.7 MB.
**SUPPLEMENTAL FILE 9**, TIFF file, 0.5 MB.
**SUPPLEMENTAL FILE 10**, TIFF file, 2.7 MB.

## ACKNOWLEDGMENTS

We thank the China Scholarship Council (CSC) for the grant of Liqiong Chen.

This research was funded by the Hungarian Government and the European Union within the frames of the Széchenyi 2020 Program (GINOP-2.3.2-15-2016-00052). The publication of this article was supported by the RRF-2.1.2-21-2022-00011 project, financed by the Government of Hungary within the framework of the Recovery and Resilience Facility.

Conceptualization: G.S., L.K., L.G.N., and C.V.; Data Curation: L.C., A.S., and S.C.; Formal Analysis: S.C., N.S., and A.S.; Funding Acquisition: C.V. and G.S.; Investigation: L.C., S.C., N.S., and A.S.; Methodology: B.P., G.S., L.K., L.G.N., G.N., G.M., and S.C.; Project Administration: G.S., L.K., L.G.N., and C.V.; Resources: G.S., L.G.N., and L.K.; Software: A.S., S.C., N.S., G.N., G.M., and L.C.; Supervision: C.V., G.S., L.G.N., and L.K.; Validation: L.K., G.S., L.G.N., and C.V.; Visualization: L.C., O.L., N.S., and S.C.; Writing – Original Draft Preparation: L.C., S.C., L.K., G.S., B.I., N.S., and L.G.N.; Writing – Review and Editing: C.V., L.C., L.G.N., S.C., G.S., O.L., N.S., and L.K. All authors have read and agreed to the published version of the manuscript.

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
