## [Reviewer comments · Microbiology Spectrum]

Microbiology Spectrum

Dual RNA-Seq profiling unveils mycoparasitic activities of *Trichoderma atroviride* against haploid *Armillaria ostoyae* in antagonistic interaction assays

Liqiong Chen, Simang Champramary, Neha Sahu, Boris Indic, Attila Szűcs, Gábor Nagy, Gergely Maróti, Bernadett Pap, Omar Languar, Csaba Vágvölgyi, László Nagy, László Kredics, and György Sipos

Corresponding Author(s): György Sipos, Soproni Egyetem

Review Timeline:

Submission Date:	November 12, 2022
Editorial Decision:	February 6, 2023
Revision Received:	April 11, 2023
Accepted:	April 11, 2023

Editor: Damian Krysan

Reviewer(s): The reviewers have opted to remain anonymous.

Transaction Report:

DOI: <https://doi.org/10.1128/spectrum.04626-22>

February 6, 2023

Prof. György Sipos
Soproni Egyetem
Faculty of Forestry
Sopron
Hungary

Re: Spectrum04626-22 (**Dual RNA-Seq profiling unveils mycoparasitic activities of *Trichoderma atroviride* against haploid *Armillaria ostoyae* in antagonistic interaction assays**)

Dear Prof. György Sipos:

I apologize for the delays in review. One of the reviewers who initially agreed could not complete the review. Instead of delaying further, we will proceed with the received review. The major issue that needs to be considered is the stringency with which the data is interpreted. I agree with the reviewer on that point and invite you to revise the manuscript.

Link Not Available

Sincerely,

Damian Krysan

Journals Department
Reviewer comments:

Reviewer #2 (Comments for the Author):

The article titled "Dual RNA-Seq profiling unveils mycoparasitic activities of *Trichoderma atroviride* against haploid *Armillaria ostoyae* in antagonistic interaction assays" by Chen and colleaues characterizes the antagonism of T.a. and the defensive response of A.o. A.o. is an ecologically devastating pathogen for new growth forests, therefore understanding the mycoparasitic effects of Ta are likely to lead to novel biocontrol methods against Ao. The authors provide well-controlled and thoughtfully executed experiment. Interpretation and subsequent analysis of the data requires a much more stringent approach.

Major Concerns

A more stringent look at what is considered a transcriptional change is likely to uncover a more specific response. As is, the analysis of the transcriptional responses seems overstated and overinterpreted. Global changes are reported but with non-standard cut-offs for expression changes which brings into question the robustness of the response. The transcriptional responses are grouped into a downtrend, metabolite, and mycoparasitic cluster. Almost none of the genes experience at least a 2-fold change in expression which is a standard cut-off. As is, much of the RNA_seq analysis is not well supported by qRT-PCR analysis. For instance, genes that were picked out from the RNA_seq analysis in figure 10 only had 4 comparisons reach statistical significance with qRT-PCR among ~20 comparisons in the figure in a set of comparisons where most should have been confirmed. I recommend the authors take a more stringent look at the transcriptional response.

Minor Concerns

Figure 1 legend needs more information to orient those not familiar with this assay or what these organisms look like

Figure 3 Why are there fold changes in the 0hr time point? Wasn't this the control that later time points were normalized to?

Figure 3 Why are most of the genes reported in each cluster between a $|0.5|$ log₂ fold change when genes in this range were filtered out according to the methods.

The cluster analyses, GO analysis, InterPro appear to be performed on all genes that fall into respective cluster trends rather than statistically different transcripts with a cutoff minimum. Non-stringent cutoffs artificially produce robust pathway enrichment as more members of the pathway become included. It's possible a global response is not descriptive of your phenotype but rather a much tighter subset of genes.

Figure 6 What is the scale of part C... fold change? Can we see statistics for the qPCR data?

Figure 9 What is the scale of part C heatmap?

Figure 13 What is the statistical test performed? How do your actin, gpd, and eIF-4 gene respond in your RNA_seq, are they stable expression? The interpretation that "All 10 genes exhibited higher expression ... indicating that our experimental results were valid" is not supported. Statistically, these higher expression levels could have been observed from random observation of identically similar populations. Most are statistically considered the same.

Staff Comments:

Preparing Revision Guidelines

Please return the manuscript within 60 days; if you cannot complete the modification within this time period, please contact me. If you do not wish to modify the manuscript and prefer to submit it to another journal, please notify me of your decision immediately so that the manuscript may be formally withdrawn from consideration by Microbiology Spectrum.

The article titled “Dual RNA-Seq profiling unveils mycoparasitic activities of *Trichoderma atroviride* against haploid *Armillaria ostoyae* in antagonistic interaction assays” by Chen and colleagues characterizes the antagonism of T.a. and the defensive response of A.o. A.o. is an ecologically devastating pathogen for new growth forests, therefore understanding the mycoparasitic effects of Ta are likely to lead to novel biocontrol methods against Ao. The authors provide well-controlled and thoughtfully executed experiment. Interpretation and subsequent analysis of the data requires a much more stringent approach.

Major Concerns

A more stringent look at what is considered a transcriptional change is likely to uncover a more specific response. As is, the analysis of the transcriptional responses seems overstated and overinterpreted. Global changes are reported but with non-standard cut-offs for expression changes which brings into question the robustness of the response. The transcriptional responses are grouped into a downtrend, metabolite, and mycoparasitic cluster. Almost none of the genes experience at least a 2-fold change in expression which is a standard cut-off. As is, much of the RNA_seq analysis is not well supported by qRT-PCR analysis. For instance, genes that were picked out from the RNA_seq analysis in figure 10 only had 4 comparisons reach statistical significance with qRT-PCR among ~20 comparisons in the figure in a set of comparisons where most should have been confirmed. I recommend the authors take a more stringent look at the transcriptional response.

Minor Concerns

Figure 1 legend needs more information to orient those not familiar with this assay or what these organisms look like

Figure 3 Why are there fold changes in the 0hr time point? Wasn't this the control that later time points were normalized to?

Figure 3 Why are most of the genes reported in each cluster between a $|0.5|$ log₂ fold change when genes in this range were filtered out according to the methods.

The cluster analyses, GO analysis, InterPro appear to be performed on all genes that fall into respective cluster trends rather than statistically different transcripts with a cutoff minimum. Non-stringent cutoffs artificially produce robust pathway enrichment as more members of the pathway become included. It's possible a global response is not descriptive of your phenotype but rather a much tighter subset of genes.

Figure 6 What is the scale of part C... fold change? Can we see statistics for the qPCR data?

Figure 9 What is the scale of part C heatmap?

Figure 13 What is the statistical test performed? How do your actin, gpd, and eIF-4 gene respond in your RNA_seq, are they stable expression? The interpretation that “All 10 genes exhibited higher expression ... indicating that our experimental results were valid” is not supported. Statistically, these higher expression levels could have been observed from random observation of identically similar populations. Most are statistically considered the same.

We thank the reviewer for carefully reviewing our manuscript and providing valuable critical comments, which proved very helpful during the preparation of the revised version. At first, we repeated all the necessary validation experiments, revisited the reviewer's remarks and suggestions, and implemented all the required corrections and changes in the text and figures.

Our responses to Reviewer #2 comments:

Reviewer #2 (Comments for the Author):

The article titled "Dual RNA-Seq profiling unveils mycoparasitic activities of *Trichoderma atroviride* against haploid *Armillaria ostoyae* in antagonistic interaction assays" by Chen and colleagues characterizes the antagonism of T.a. and the defensive response of A.o. A.o. is an ecologically devastating pathogen for new growth forests, therefore understanding the mycoparasitic effects of Ta are likely to lead to novel biocontrol methods against Ao. The authors provide well-controlled and thoughtfully executed experiment. Interpretation and subsequent analysis of the data requires a much more stringent approach.

Answer:

Thank you for your careful critical remarks, and we appreciate your interest in our study.

Reviewer #2 / Major Concerns

"A more stringent look at what is considered a transcriptional change is likely to uncover a more specific response. As is, the analysis of the transcriptional responses seems overstated and overinterpreted. Global changes are reported but with non-standard cut-offs for expression changes which brings into question the robustness of the response. The transcriptional responses are grouped into a downtrend, metabolite, and mycoparasitic cluster. Almost none of the genes experience at least a 2-fold change in expression which is a standard cut-off. As is, much of the RNA_seq analysis is not well supported by qRT-PCR analysis. For instance, genes that were picked out from the RNA_seq analysis in figure 10 only had 4 comparisons reach statistical significance with qRT-PCR among ~20 comparisons in the figure in a set of comparisons where most should have been confirmed. I recommend the authors take a more stringent look at the transcriptional response."

Answer:

First of all, we apologize for the accidentally mislabelled Figure 3, where we indicated log2fc on the Y-axis instead of z-score! Unfortunately, it was misleading from the very beginning.

In fact, as described in the methodology section, we applied a rigorous approach to identify candidate genes for the metabolite and mycoparasitic phase-specific and the "downstream" analyses. Specifically, we only included "downtrend" genes with a p-value less than 0.05 and a minimum log2 fold change of 1 or above between the actual time points analyzed. We chose these criteria to ensure that the identified genes had a significant and biologically meaningful change in their expression between the two conditions compared.

According to the reviewer's comments, we wrote a new methodology subsection titled "Time course analysis" (lanes: 598-625 / Marked up pdf), where we describe all details of the differential gene expression analysis, the identification of the significantly upregulated genes, and the calculation of z-scores for Figure 3.

After observing a robust initial exchange between the mycoparasite and its prey, including an immediate upregulation of several apoptotic genes during the metabolite stage, we carefully chose the genes for the qRT-PCR validation. The list of genes from *Armillaria* enclosed biosynthetic genes from the kynurenine pathway producing quinolinic acid (QA), the PK genes containing the SnoaL domain

involved in the biosynthesis of polyketide antibiotics, and three representative markers of the apoptotic pathways. Regarding the QA biosynthetic pathway, we only included three genes from the initial steps, as we aim to study this process in more details and report the results in a follow-up publication. Our repeated qPCR experiments confirmed the significant upregulation (lanes: 352 - 370 and 668 - 676 / Marked up pdf) of all the selected 8 AO and 3 TA genes (after including an additional TA gene).

Concerning the robustness of the interaction between TA and AO, we already observed an "airborne"-based communication at the gene expression levels between the fungal partners during their control face-off after growing them on separate plates. Such initial molecular responses moved towards an intense molecular "exchange" during the first "soilborne" metabolite phase. Here we have parallel studies also focused on identifying the volatile effectors and aiming at a separate publication.

Reviewer #2 / Minor Concerns

Figure 1 legend needs more information to orient those not familiar with this assay or what these organisms look like

Answer:

According to the helpful suggestion of the Reviewer the figure legend of Figure 1 was modified as follows (lanes: 987-994 / Marked up pdf):

Fig 1. Antagonistic effect of *T. atroviride* (TA) against various diploid (D1-D4) and haploid (H1-H4) strains of *A. ostoyae* (AO) in dual culture assay on PDA medium. AO and TA strains were inoculated to the right and left side of the plates, respectively, as described in section 4.2. Melanized rhizomorphs were developed on the surface of the medium in the case of all diploid AO strains, while haploid AO strains did not produce any rhizomorphs and were easily overgrown by TA, suggesting that they are more suitable for studying pure hyphal interactions of the two fungi and the mycoparasitic attack of TA on AO.

Reviewer #2

Figure 3 Why are there fold changes in the 0hr time point? Wasn't this the control that later time points were normalized to?

Figure 3 Why are most of the genes reported in each cluster between a $|0.5|$ log₂ fold change when genes in this range were filtered out according to the methods.

Answer:

Regarding Figure 3, as we have already indicated above, the Y-axis was accidentally mislabelled indicating log₂fc instead of z-score. We corrected and submitted the new revised Figure 3. We hope that considering the z-score data explains all of the critically mentioned details.

Reviewer #2

The cluster analyses, GO analysis, InterPro appear to be performed on all genes that fall into respective cluster trends rather than statistically different transcripts with a cutoff minimum. Non-stringent cutoffs artificially produce robust pathway enrichment as more members of the pathway become included. It's possible a global response is not descriptive of your phenotype but rather a much tighter subset of genes.

Answer:

According to the Reviewer's valuable major concerns and comments, as already mentioned above, we included a new subsection describing all details in identifying significantly affected clusters with reliable GO and InterPro profiles. Therein, in addition to rigorously identifying candidate genes, we used a p-value cutoff of 0.05 for our enrichment analyses, to further ensure that the identified pathways were statistically significant (lanes: 615-622 / Marked up pdf).

Reviewer #2

Figure 6 What is the scale of part C... fold change? Can we see statistics for the qPCR data?

Answer:

We used the 'scale ()' function in R to standardize the TMM normalized expression counts in our study. This standardization made it easier to compare data across different samples, as the values were in standardized scale. It was then used to plot the heatmap.

Regarding the qPCR data, as already mentioned above, we tested three genes involved in the initial steps of QA biosynthesis and the qPCR results confirmed that they were significantly upregulated in both metabolic and mycoparasitic stages (lanes: 360-362 / Marked up pdf). We have also submitted an updated Figure 6.

Reviewer #2

Figure 9 What is the scale of part C heatmap?

Answer:

Thank you for your comment. According to the question the following details were included in the methods section (lanes: 623 -625 / Marked up pdf): Regarding Figure 9, the scales were generated using the z-scores, calculated by the heatmap.2 function from the gplots package in R. For the input file, the averaged expression values were used.

Reviewer #2

Figure 13 What is the statistical test performed? How do your actin, gpd, and eIF-4 gene respond in your RNA_seq, are they stable expression? The interpretation that "All 10 genes exhibited higher expression ... indicating that our experimental results were valid" is not supported. Statistically, these higher expression levels could have been observed from random observation of identically similar populations. Most are statistically considered the same.

Answer:

Thank you very much for your comment. We have repeated the qRT-PCR experiments. All measurements were performed in two technical and at least three biological replicates. Significance was calculated with paired t-test using the GraphPad Prism 7.00 program (GraphPad Software, La Jolla, CA, USA). P values less than 0.05 were considered as statistically significant (lanes: 672-676 / Marked up pdf). RNAseq analysis did not show significant changes in the expression of "housekeeping" genes (e.g. actin, gpd, and eIF-4). Actin, gpd, and eIF-4 genes showed constitutive expression levels under the tested conditions. Although the relative transcript levels of actin and gpd slightly decreased in the Mycoparasitic stage, which can be explained by the fact that a strong cell death process is initiated in the *Armillaria* in this stage. We have submitted an updated Figure 13.

April 11, 2023

Prof. György Sipos
Soproni Egyetem
Faculty of Forestry
Sopron
Hungary

Re: Spectrum04626-22R1 (**Dual RNA-Seq profiling unveils mycoparasitic activities of *Trichoderma atroviride* against haploid *Armillaria ostoyae* in antagonistic interaction assays**)

Dear Prof. György Sipos:

Thank you for revising the manuscript. You have addressed the reviewer's concerns and I am pleased to accept the manuscript.

Your manuscript has been accepted, and I am forwarding it to the ASM Journals Department for publication. You will be notified when your proofs are ready to be viewed.

Sincerely,

Damian Krysan
Editor, Microbiology Spectrum
